# An Automated Hardware-in-Loop Testbed for Evaluating Hemorrhagic Shock Resuscitation Controllers

**DOI:** 10.3390/bioengineering9080373

**Published:** 2022-08-07

**Authors:** Eric. J. Snider, David Berard, Saul J. Vega, Sofia I. Hernandez Torres, Guy Avital, Emily N. Boice

**Affiliations:** 1U.S. Army Institute of Surgical Research, JBSA Fort Sam Houston, San Antonio, TX 78234, USA; 2Trauma and Combat Medicine Branch, Surgeon General’s Headquarters, Israel Defense Forces, Ramat-Gan 52620, Israel; 3Division of Anesthesia, Intensive Care and Pain Management, Tel-Aviv Sourasky Medical Center, Tel-Aviv 64239, Israel

**Keywords:** fluid resuscitation, infusion, controllers, hemorrhage, hardware-in-loop, flow loop, closed-loop, test platform

## Abstract

Hemorrhage remains a leading cause of death, with early goal-directed fluid resuscitation being a pillar of mortality prevention. While closed-loop resuscitation can potentially benefit this effort, development of these systems is resource-intensive, making it a challenge to compare infusion controllers and respective hardware within a range of physiologically relevant hemorrhage scenarios. Here, we present a hardware-in-loop automated testbed for resuscitation controllers (HATRC) that provides a simple yet robust methodology to evaluate controllers. HATRC is a flow-loop benchtop system comprised of multiple PhysioVessels which mimic pressure-volume responsiveness for different resuscitation infusates. Subject variability and infusate switching were integrated for more complex testing. Further, HATRC can modulate fluidic resistance to mimic arterial resistance changes after vasopressor administration. Finally, all outflow rates are computer-controlled, with rules to dictate hemorrhage, clotting, and urine rates. Using HATRC, we evaluated a decision-table controller at two sampling rates with different hemorrhage scenarios. HATRC allows quantification of twelve performance metrics for each controller configuration and scenario, producing heterogeneous results and highlighting the need for controller evaluation with multiple hemorrhage scenarios. In conclusion, HATRC can be used to evaluate closed-loop controllers through user-defined hemorrhage scenarios while rating their performance. Extensive controller troubleshooting using HATRC can accelerate product development and subsequent translation.

## 1. Introduction

Hemorrhage is the leading cause of preventable death in both civilian [1] and military [2] casualties. In the military setting, hemorrhage is the leading cause of death in casualties that have arrived alive to a military treatment facility, with their death attributed mostly to the effects of hemorrhagic shock [3]. The loss of blood volume leads to impaired oxygen delivery to the tissues (DO_2_), creating a burden of oxygen debt that impairs the function of vital body systems [4], eventually resulting in ischemic tissue damage and ultimately death. The key to mitigating and reversing this process is through early goal-directed restoration of DO_2_ while avoiding any increase in blood pressure, as this may cause resumption or exacerbation of the bleeding. This is especially true in remote environments where hemorrhage control is not always achievable, a concept known as Remote Damage Control Resuscitation (RDCR) [5].

In conjunction with hemorrhage control, fluid therapy remains a cornerstone of treating blood loss (hypovolemia), although the nonsystematic use of endpoints to guide resuscitation results in conflicting outcomes [6]. Hypovolemia can result in damage, dysfunction, and failure of both tissues and end organs. Precise fluid management is favored, and has been classically managed by several teaching manuals and clinical practice guidelines [7,8]. These approaches are based on a fixed formula-based regimen that replaces fluid (blood products, crystalloids, colloids, etc.) to compensate for blood loss.

Alas, providing accurate goal-directed fluid therapy (GDFT) can be challenging, requiring frequent attention and a certain amount of expertise. Extreme circumstances, such as mass casualty incidents or large-scale military operations with large numbers of casualties and prolonged casualty care, may hinder the availability of such expertise and attention. Automation of GDFT protocols within medical devices has the potential to improve their performance while decreasing task loads, optimizing utilization of precious resources, especially blood products.

Automated resuscitation is a large research area, with multiple approaches being taken to achieve GDFT. In this type of therapy, an automated controller directs infusion rates and times, which are adapted in response to the measured value of a specific physiologic input, e.g., blood pressure. Algorithms that analyze endpoint data and continuously adapt fluid infusion rates through automated infusion pumps, have been tested in silico [9], as well as in vivo with various animal and human models [10,11], using a variety of commonly used resuscitation fluids [12]. These closed-loop controllers range from medical devices containing simple decision tables [11,13] to more sophisticated models such as fuzzy logic [11,14,15], proportional integral derivative [16,17], and adaptive controllers [9,16,18,19].

We previously developed a benchtop flow loop testing platform utilizing two PhysioVessels [20] to bridge the gap between in silico and animal testing. In the present study, we further the development of a Hardware-in-Loop Automated Testbed for Resuscitation Controllers (HATRC) for fully automated and robust device testing in lieu of extensive animal testing. These modifications incorporate complex bleed profiles, increased subject variability, and allow for head-to-head comparisons of different controllers. Additionally, HATRC enables the assessment of resuscitation controllers using a comprehensive list of standardized performance metrics to evaluate the performance of different controller designs, types, and tuning.

## 2. Materials and Methods

### 2.1. Overview of Hardware-in-Loop Automated Testbed for Resuscitation Controllers

We constructed HATRC to replicate the physiological arterial pressure response of a swine model to bolus infusions of either whole blood (WB) [21] or crystalloid fluids, while simulating various hemorrhage and resuscitation scenarios (Figure 1). As a proof-of-concept, the working fluid was tap water, although the system should operate similarly with any low viscosity non-compressible fluid. Circulation in the loop was driven by a SuperPump Pulsatile Pump (ViVitro Labs, Victoria, BC, Canada), labelled “SP” in Figure 1, with tunable stroke volume and pulse rates. Due to the relatively low total volume of the system and to keep pressures within a physiological range, a 2 mL stroke volume was used at 120 BPM. After the pump, a three-way valve (TwV1, Figure 1) (Cole Parmer, Vernon Hills, IL, USA) attached an arterial line tube leading to a pressure transducer (PT, Figure 1) (ICU Medical, San Clemente, CA, USA) connected to a patient monitor (PM, Figure 1) (Infinity Delta XL, Draeger, Lubeck, Germany) for monitoring the arterial pressure wave data in the system. Downstream from the connection site of the PT was a second three-way valve (TwV2, Figure 1) that bifurcated the main loop and directed the flow to follow only one path at a time. One branch was unaltered, allowing unobstructed flow, while the alternate branch included an adjustable regulating valve (RV, Figure 1) to increase pressure by increasing fluidic resistance, similar in effect to systemic vascular resistance increased by vasopressor administration [22,23]. While the effect size on pressure can be adjusted, an approximate 15 mmHg step was set to mimic clinical effect magnitudes [23].

Next, the two branches merged back into a single path followed by a tee fitting connected to a third three-way valve (TwV3, Figure 1). This valve allowed a single connection point for two different PhysioVessels, one designed to mimic a WB pressure response (PV_WB_, Figure 1) and one for a crystalloid pressure response (PV_Crys_, Figure 1). PhysioVessels were previously developed as a hydrostatic pressure reservoir to provide the equivalence of venous capacitance and filling pressure to the system [20]. The PhysioVessels were elevated above the rest of the loop to produce an average systemic pressure reading at PT of 68 mmHg when filled. After passing through the integration site of the PhysioVessels, the loop was then closed at the inlet to the SuperPump.

An additional set of components external to the main loop included two peristaltic pumps (Masterflex L/S, Masterflex Bioprocessing, Vernon Hills, IL, USA); one was used as an infusion pump (IP, Figure 1) to supply volume to the system, and the other as an outflow pump (OP, Figure 1) to simulate the basal urinary output as well as a variety of hemorrhage rates (see “HATRC Experimental Setup” section). The outlet tube of IP was joined with the inlet tube of OP at another tee fitting leading to a fourth three-way valve (TwV4, Figure 1). At this valve, two tubes connected to ports on either of the PhysioVessels, enabling both resuscitation and outflow to be directed to either vessel as desired between or during infusion tests. The positioning of the connections for these two pumps was chosen to allow the addition or removal of volume from the system, thus producing a change in pressure while not disturbing the flow within the loop itself, as this could result in undesirable pressure perturbations. Infusion and outflow pump rates and the timing of their operations were controlled by a computer (PC, Figure 1) through a standard serial port using MATLAB R2021b (MathWorks, Natick, MA, USA).

### 2.2. PhysioVessel Modifications

In our previous work, we introduced the methodology and design process for the PhysioVessel [20]. Briefly, the PhysioVessel was developed to successfully reproduce normalized pressure–volume responsiveness as seen in a swine resuscitation model [24]. This was accomplished by finding the least squares regression curve fits for normalized mean arterial pressure (MAP) vs. infused volume for WB and crystalloid bolus resuscitations. By solving these equations, volume as a function of pressure could be identified for each infusate’s physiological response. By applying the fluid dynamic principle that directly correlates the height of a column of fluid above a point to the resulting hydrostatic pressure at that point, the volume of fluid in a vessel was defined as a function of its height, allowing hydrostatic pressure to be substituted for height and resulting in the volume likewise being defined as a function of the pressure for the PhysioVessel model. In the case of resuscitation with WB, the physiological response was linear, and the resulting vessel geometry was a right cylinder. On the other hand, crystalloid was found to have a parabolic response, and resulted in a vessel resembling a funnel with curved walls. Previously, we evaluated the average of 4–5 datasets of each infusate that contained calculated MAP values sampled once every 5 s, obtained from a Millar Mikro-Tip pressure transducer-tipped catheter (Millar Instruments, Inc., Houston, TX, USA). Datasets were normalized for pressure and volume by dividing each data point by the maximum corresponding value observed within that data set.

For the current work, we expanded our analysis to include hemorrhage resuscitation data from twelve WB and nine crystalloid-infused swine collected from the same experimental dataset [20,24]. For a more refined analysis, raw pressure data sampled at 500 Hz were used for the current analysis. A moving average window with a one-second width was used to smooth the data. Swine resuscitation data for each infusate type were averaged and their corresponding regression curves were generated. As opposed to the original PhysioVessel development methodology [20], the current method of analysis foregoes any normalization of the data to maximum or minimum values. However, the infusion rates used in the original experiments were normalized by the mass (in kg) of each pig, provided as 2.8 mL × min^−1^ × kg^−1^ for WB and 4.5 mL × min^−1^ × kg^−1^ for crystalloid. This preserved the ability of our system to be scaled to volumes below physiological ranges while reliably mimicking the pressure response.

To produce the right cylinder for PV_WB_, we used a 3-inch (76 mm) diameter PVC pipe (McMaster-Carr, Elmhurst, IL, USA) modified to have an equivalent radius of 27 mm. A base was constructed using a 3-inch PVC end cap and ports were added near the bottom to attach the tubes leading to the loop and infusion/outflow pumps. PV_Crys_ was modeled using computer-aided-design software (Solidworks, Waltham, MA, USA) and 3D printed (Raise3D, Irvine, CA, USA). To compare the pressure–volume responses of the new PhysioVessel models, the hydrostatic pressure was set at approximately 40 mmHg. Water was infused at a constant rate of 500 mL/min into the vessel and pressure was recorded over time. The circulating pump of HATRC was not operating during these tests. Two patient variabilities were added to each vessel, functionally changing the equivalent radius of the PV used. Two WB variations were developed, while the original PV_Crys_ model [20] was used with height adjustments to achieve two crystalloid variations.

Resuscitation of critically injured patients may be required in austere and resource-limited settings. As such, HATRC can simulate infusion of a limited volume of WB, then switch to crystalloid. TwV3 (Figure 1) dictated whether the static pressure of the flow loop was supplied by PV_WB_ or PV_Crys_, while TwV4 directed the infusion/outflow of the pumps to which vessel was in use. This capability was demonstrated using three starting volumes of WB infusate (300 mL, 600 mL, and 900 mL), followed by crystalloid infusate to a target MAP of 68 mmHg. To do this, both PVs were drained and PV_WB_ was filled to supply a pressure of 40 mmHg prior to the start of infusion. Infusion of WB was supplied at 500 mL/min until the specified volume was reached, at which point the infusion was paused, PV_Crys_ was filled to reach an equivalent pressure to PV_WB_, both TwV2 and TwV3 were switched to PV_Crys_, and infusion was resumed at 500 mL/min until a MAP of 68 mmHg was reached.

### 2.3. HATRC Experimental Setup

Closed-loop resuscitation controller experiments were performed using a computer script written in MATLAB (Figure 2A). Each simulated resuscitation scenario began by initializing the system to a starting pressure of 40 mmHg with a corresponding system volume loss of ~2300 mL. During simulation, an outflow rate was set via OP and a fluid-resuscitation controller was allowed to set an infusion rate via IP until the target MAP was reached. After reaching 99% of the target MAP, an additional 15 min was allowed for the controller to stabilize prior to moving to the next simulated scenario. For scenario four described below, the total scenario time was limited to 30 min regardless of whether the target MAP was reached or not.

Four different hemorrhage and resuscitation scenarios were simulated using HATRC (Figure 2B). These scenarios were coordinated to run in succession by operating OP at dynamically calculated rates to account for evolving conditions during the simulation and pausing at predetermined time points to allow for manual operation of the valves. All four test scenarios challenged the ability of the fluid-resuscitation controller under evaluation to achieve and maintain a target MAP of 68 mmHg. A data acquisition system (PowerLab, ADInstruments, Sydney, Australia) was used to capture incoming arterial pressure readings from the patient monitor as a raw waveform and stream it to MATLAB in real-time using a MATLAB function developed by AD Instruments. The simulation script then calculated a MAP as the average value of the raw waveform in a five second sampling window and used it as the input for the controller.

### 2.4. Hemorrhage Test Scenarios

The first test scenario in the sequence was a simple fluid resuscitation protocol with whole blood from an initial MAP of 40 mmHg. The second scenario simulated a large re-bleed (e.g., an extremity tourniquet suddenly coming loose) followed again by a resuscitation with WB. The third scenario repeated the re-bleed and WB resuscitation from scenario two with an additional 15-mmHg step increase in MAP, as can be produced via vasopressor administration, immediately following the re-bleed. After a period of 10 min, the pressure increase was removed and the controller’s response to the sudden drop in MAP was recorded. Finally, the fourth test scenario in the sequence simulated a fluid resuscitation protocol using a crystalloid infusate, and Trauma Induced Coagulopathy (TIC) was simulated [25] (Figure 2B).

Throughout all the test scenarios, the computer script managed a simulated urine output fixed at 5 mL/min when the MAP was 50 mmHg or higher, mimicking a basal urine rate, and zero when the MAP dropped below this threshold. While the urine rate selected is above the upper limit of physiological urinary flow rates observed in swine of 2.03 +/− 1 mL/kg/h [26], the scenarios and testing setup can be adjusted to meet the needs of end users. The higher rate was selected due to peristaltic pump hardware limitations with slower flow rates.

The calculations for the hemorrhage component of the outflow rates during the test scenarios were influenced mainly by MAP and a time-varying “hemorrhage factor” (HF). In essence, at any given time, the hemorrhage rate was the mathematical product of the system’s MAP and HF (Figure 2C), which was then varied by ±5% as a way to simulate noise. Around the value of HF, the rate was constantly re-calculated throughout each scenario while being constrained to a range able to produce bleed rates appropriate for each simulation and the hardware constraints of OP. The initial value of HF was set according to the scenario being simulated as one of two possible hemorrhage levels; one produced a bleed rate of 120 mL/min at a MAP of 68 mmHg, while the other produced a lower rate of 60 mL/min at the same pressure. During the simulation, without any external influence HF was gradually reduced in magnitude over time by an infusate-dependent coagulation factor, thus mimicking normal blood coagulation and reducing the hemorrhage rate [27]. However, this coagulation process could be overridden by evolving conditions during the HATRC simulation affecting the MAP, or by pre-programmed events.

The calculation of HF could be affected by MAP falling outside of a pre-determined range. On the low end of this range, for a MAP below 30 mmHg, hemorrhage rates were assumed to become negligible, and HF was set to zero. Conversely, should the infusion controller ever drive MAP above 70 mmHg, the value of HF was gradually increased to produce a higher hemorrhage rate for as long as the pressure was above that threshold. This was to simulate internal bleeding caused by blood clots becoming dislodged due to increased pressure [28].

Events programmed in HATRC scenarios could override the normal calculation of HF. A sudden massive hemorrhage scripted into the simulation, such as the loose tourniquet example in the second scenario, would temporarily and immediately set HF to a high value that resulted in a hemorrhage of 255 mL/min at a MAP of 68 mmHg (a rate similar in magnitude to resting femoral artery blood flow in humans, approximately 280 mL/min [29]). In another scripted event in scenario four, HF would be gradually increased to produce higher bleeding rates up to a user-defined limit, thus preventing the simulated coagulation process from ever occurring.

### 2.5. Closed-Loop Resuscitation

For controlling infusion, a decision table controller previously developed by Marques et al., 2017 was adapted for use as a case study of how well medical devices such as closed-loop controllers could be evaluated with HATRC [11]. The controller was scaled for relative MAP values and flow rates were adjusted to align with maximum flow rates for the infusion pump. The decision table logic set the flow rate based on distance from the target MAP using six logical steps to slow perfusion as target MAP was reached, thus avoiding overshooting (Figure 2D). The initial logic by Marques et al., 2017 was developed for use with in vivo animal studies and was set to make adjustments every 120 s [11]. In order to highlight differences in the respective controllers’ performance, two controller sample rates were evaluated: 120 and 5 s.

### 2.6. Performance Calculations

HATRC can be used in conjunction with several performance metrics to compare the performance of the controller during simulated hemorrhage and resuscitation scenarios (Figure 3). Among the performance metrics used were those proposed by Varvel et al., 1992 for assessing closed-loop controllers in clinical applications, all based on their definition of Performance Error (PE), as shown in Equation (1) [30].
(1)$${\mathrm{PE}}_{i}=\frac{P_{i}-P_{\mathrm{Target}}}{P_{\mathrm{Target}}}\times 100$$
where P_i_ is the *i*th measurement of the mean arterial pressure and P_Target_ is the desired target pressure (i.e., the controller’s setpoint). Four measurements are then derived from these Performance Errors, as shown in Equations (2)–(5).
(2)$$\mathrm{Median}\;\mathrm{Performance}\;\mathrm{Error}\;\mathrm{or}\;\mathrm{MDPE}=\mathrm{median}\left\{{\mathrm{PE}}_{i},i=1\ldots ,N\right\}$$
where N is the total count of Performance Error measurements.
(3)$$\mathrm{Median}\;\mathrm{Absolute}\;\mathrm{Performance}\;\mathrm{Error}\;\mathrm{or}\;\mathrm{MDAPE}=\mathrm{median}\left\{\left|{\mathrm{PE}}_{i}\right|,i=1\ldots ,N\right\}$$
(4)$$\mathrm{Wobble}=\mathrm{median}\left\{\left|{\mathrm{PE}}_{i}-\mathrm{MDPE}\right|,i=1\ldots ,N\right\}$$
(5)$$\mathrm{Divergence}=60\times \frac{\sum_{i=1}^{N}\left|{\mathrm{PE}}_{i}\right|\times t_{i}-\left(\sum_{i=1}^{N}\left|{\mathrm{PE}}_{i}\right|\times \sum_{i=1}^{N}t_{i}\right)÷N}{\sum_{i=1}^{N}t_{i}^{2}-{\left(\sum_{i=1}^{N}t_{i}\right)}^{2}÷N}$$
where N is the total count of Performance Error measurements, t_i_ is the time in minutes for the corresponding PE_i_, and the factor of 60 is used to represent the measurement of divergence in units of percentage per hour.

The remaining controller performance metrics calculated were:

Relative Overshoot

This is the maximum arterial pressure measured at any point in time relative to the steady-state pressure (Figure 3A). In turn, the steady-state pressure is the value after which subsequent pressure measurements do not deviate by more than ±5% [31,32].

Effectiveness

Simply stated, this is the percentage of time that the controller was able to maintain MAP within ±5 mmHg of the target pressure (Figure 3A) [11].

Efficiency

The amount of time required for the controller to raise MAP from its initial measurement to 90% of its steady-state value, akin to the commonly used controller metric of rise time (Figure 3A) [31,32].

Volume Efficiency

A modification of the ‘Efficiency’ definition from Marques et al., 2017 [11], this is a simple ratio of the volume infused by the controller to the volume lost or removed from the system (outflow).

Average Infusion Rate

Calculated as the arithmetic mean of the rates at which the system was infused by the controller.

Area to Setpoint

Finally, we propose in this work a pair of new metrics intended to represent the clinical impact of the controller error over the period of time during which said controller performs fluid resuscitation. These new values, collectively termed as “Area to Setpoint” (and individually as Area_above_ and Area_below_), are easily identified in the graph of MAP vs. time as the shaded areas between the pressure plot and the system’s target pressure (Figure 3C); they are meant to quantify over- and under-resuscitation, respectively, as a cumulative burden throughout the testing period. Area_below_ represents global oxygen debt beyond the accepted physiological debt. Area_above_ represents over-resuscitation comparing to DCR goals, which in theory correlates to increased blood loss, leading to less optimal utilization of resources such as blood products. The new metrics are calculated as shown in Equations (6) and (7):(6)$${\mathrm{Area}}_{\mathrm{above}}=\frac{\sum_{i=2}^{N}\left(P_{i}-P_{\mathrm{Target}}\right)\times \left(t_{i}-t_{i-1}\right)}{P_{\mathrm{Target}}},\;\mathrm{for}\;\mathrm{all}\;P_{i}>P_{\mathrm{Target}}$$
and
(7)$${\mathrm{Area}}_{\mathrm{below}}=\frac{\sum_{i=2}^{N}\left(P_{i}-P_{\mathrm{Target}}\right)\times \left(t_{i}-t_{i-1}\right)}{P_{\mathrm{Target}}},\;\mathrm{for}\;\mathrm{all}\;P_{i}<P_{\mathrm{Target}}$$
where N is the total number of pressure measurements and P_i_, P_Target_, and t_i_ are as previously described. As the areas are normalized by the target pressure, the units of this measurement are provided in minutes.

Another related area value was calculated, namely, Rise Area to Setpoint. This metric involves the same calculation for Area_below_ as just described, except that N is instead fixed such that $P_{N}=0.9\times P_{\mathrm{Target}}$. This alternative area measurement was calculated for the first resuscitation scenario in order to indirectly represent the oxygen debt resulting from each controller’s initial resuscitation profile from a set starting point that was uniform among all tested controller configurations, thus serving as an improvement over rise time. Only the first scenario was evaluated, as the scenarios occurred in succession, thus subsequent calculations for this metric are influenced by the previous scenario. 

## 3. Results

### 3.1. PhysioVessel Modification and Modeling Subject Variability

The pressure response to infused WB volume in the swine model produced a right cylinder with an equivalent radius of 27 mm, while the crystalloid response resulted in a vessel in which the radius changed as a function of the height, $r=\sqrt{\frac{25}{\pi 8.1855*\sqrt{39.5608-z}}}$, where r is the radius and z is the height. Comparisons of both PhysioVessels to the physiological results are shown in Figure 4. A standard run of the base vessel is shown alongside the averaged swine dataset with an overlay of the regression curve line and a shaded region covering ±1 standard deviation (SD). Both PhysioVessel types produced pressure responses closely correlated to the results of the swine model, with the slope of the WB and shape of the crystalloid plots following the swine data. Only a slight offset was observed due to operating at slightly higher pressures than the previously performed animal experiments. We included an additional two controlled variations for each vessel type in order to demonstrate the system’s ability to incorporate patient variability into the pressure response profile. These fell within the ±1 SD range of the animal data as well. An additional testing capability for HATRC is the ability to switch between PhysioVessels to mimic switching of infusate types. This is highlighted by the change from WB to crystalloid after different volumes, which mimics resource-limited resuscitation (Figure 4C). The switch-over points on the plots are marked for easy identification of the transition point. As anticipated, the plots follow a linear response during the WB infusion, then convert to a parabolic response after switching to crystalloid.

### 3.2. Controller Evaluation Using the Automated Bleed Logic Test Platform

#### 3.2.1. Overall Testing Scenario Results for a Decision Table Controller

To highlight the utility of this test platform, we evaluated two hemorrhage severities (high and low) with a decision table resuscitation controller adapted from Marques et al., 2017, with two sampling rates of 120 s and 5 s [11]. Four testing scenarios were connected in series to enable evaluation of controller design across the four events in one test run: (i) an initial blood loss resuscitated by WB, a rapid blood loss similar to tourniquet failure resuscitated by WB; (ii) without and (iii) with bolus vasopressor delivery; and (iv) resuscitation with crystalloid with an active hemorrhage.

Comparing controller sampling rates in a high bleed rate scenario, the effect of sampling rate was pronounced (Figure 5). At a slower 120 s sampling rate, the controller overshot the pressure target, which triggered a clot burst, mimicking re-bleed in the test platform that occurred in each testing scenario. This was not observed with the 5 s sampling rate, highlighting the need to tune the sampling rate of the infusion controller appropriately for the process at hand. Regardless of the overshoot challenge, both controller setups were able to reach target pressure for each testing scenario, including after a large pressure drop simulating a bolus vasopressor rapidly wearing off, as well as when resuscitating with crystalloid, which has a drastically different pressure responsiveness than PV_WB_.

Figure 6 shows the results of the controller comparison at an increased hemorrhage rate (“high”). As expected, the higher bleed rates required higher infusion amounts during the initial hemorrhage scenario, although the controller continued to overshoot and trigger re-bleed in the test platform at the 120 s sampling rate. The most pronounced difference produced by switching to a more rapid bleed rate was seen when resuscitating against coagulopathy effects with crystalloid (Test Scenario #4). This scenario paired a faster bleed rate with the lower volume responsiveness nature of crystalloid infusion, resulting in failure to reach the pressure target. Neither controller configuration was able to achieve a steady state during the test period, as both were observed diverging from the set point due to the hemorrhage remaining greater than the infusion rate. In summary, a wide range of testing scenarios can be configured with the designed HATRC platform, allowing for more robust controller testing during in vitro benchtop studies.

#### 3.2.2. Performance Metric Evaluation

Lastly, we highlight how HATRC and its various testing scenarios can be utilized for calculating key control systems’ performance metrics. Calculations for each of twelve metrics are detailed in the methodology. Results for each testing scenario and across the entire test region are shown in the Appendix A. Overall, the various testing scenarios of HATRC result in heterogeneous performance for the controller configurations. The median performance error was lower on average for the 5 s sampling rate setup compared to the 120 s sampling rate (Figure 7A). High hemorrhage during Testing Scenario #4, where a constant bleed and crystalloid infusate was used, resulted in high MDPE values, potentially indicating that the decision table logic was not sufficient for these scenarios. Volume efficiency, a ratio of volume infused to volume lost, highlights which controller configuration was best able to conserve fluid. This metric was heavily influenced by the repeated pressure overshoot and re-bleeds that were triggered at the slower 120 s sampling rate (Figure 7B). Wobble after reaching steady state was influenced by the 120 s sampling rate overshooting and re-bleeding as well (Figure 7C). The quicker sampling rate was much more stable, while the slower sampling times led to a decreased wobble value for the 5 s controller configuration. In addition, during high hemorrhage in Testing Scenario #4, the controller drifted from the steady state, an effect captured by the wobble metric (Figure 7C). Another parameter we introduced to evaluate controller performance was the area below the target pressure, which represents a means of measuring the total hypovolemic burden caused by the controller responsiveness. As expected, the total hypovolemic burden was larger for high hemorrhage rate scenarios when compared to low hemorrhage rates (Figure 7D). While the scope of the present study is to highlight the test platform and how it can be used to evaluate controllers against a range of testing situations, at this time only data for a single controller are shown. However, a standardized platform such as this can be instrumental in comparing performance both between controllers with different tuning setups and between entirely different controller types.

## 4. Discussion

Hemorrhage remains a leading cause of preventable death in both civilian and military medicine. In austere and battlefield trauma situations medical expertise is often limited, and the expertise needed to perform fluid resuscitation on casualties may be lacking. This is further exacerbated during mass casualty scenarios, in which medical personnel are overburdened and cannot properly manage resuscitation of multiple casualties. As a result, automated fluid resuscitation controllers are being developed with different degrees of automation and decision engines based on clinical or animal fluid therapy datasets [33]. Troubleshooting these controllers for optimal performance and subsequent comparison between controller logics can be a time- and resource-intensive task. Here, we have presented HATRC, a benchtop flow loop platform with automated complex hemorrhage scenarios for troubleshooting and evaluating hemorrhage resuscitation controllers and hardware.

The resuscitation controller test platform developed here is unique from other methods [31] in that it allows for hardware-in-the-loop testing of controllers in a laboratory test bed that physically represents the pressure–volume relationship. In addition, arterial pressure sensors record system pressure in real-time and can be sampled at the rates required for proper controller input. HATRC uses physical sensors instead of the simulated data relied on by in silico simulations, as the latter are subject to electronic interference/noise, allowing HATRC to account for other real-world factors that computer simulations may fail to consider. A physical infusion pump in the loop allows for real-time adjustment by the controller output. For physiological relevancy, the pressure–volume dynamics of the system are specified by two PhysioVessel models to mimic swine responsiveness during hemorrhage resuscitation. As fluid responsiveness is patient-dependent, with a few simple design changes the PhysioVessel system shown here is able to create the different pressure–volume dynamics required to better account for subject variability when designing and testing resuscitation systems. While benchtop controller evaluation cannot replace in vivo testing, it can precede or supplement animal testing in order to narrow down the selection of controller architectures or conduct initial troubleshooting, which can reduce the animal burden [17,34]. The design process for fabricating PhysioVessels is simple, and can be tuned to the end user’s test needs or combined with multiple PhysioVessel shapes to replicate hemorrhage scenarios in which infusate types may need to be swapped, for example due to limited whole blood resources. Another feature of HATRC is an immediate change in fluidic resistance affecting system pressure, which can mimic vasopressor administration. As presented, a pressure increase of 15 mmHg was immediate and remained for 10 min, similar to a vasopressor response [23]. As this is incorporated as a vasculature resistance increase, magnitude and duration can be tuned to the end user’s needs or integrated with automated needle valve adjustments to allow for closed-loop vasopressor controller evaluation.

Another key feature of the presented benchtop hemorrhage test platform is automated outflow logic, which together with the effect of the infusate type on capillary interstitial fluid transfer results in a flexible system for which a limitless number of potential hemorrhage scenarios can be devised [17]. Interstitial fluid transfer from the vasculature was accounted for here with whole-blood and crystalloid PhysioVessel designs [20]. Additionally, the outflow hemorrhage and urine logic are responsible for driving an outflow pump under automated computer control. If HATRC pressure is too low, hemorrhage and urine rates are halted. The hemorrhage rate was set up to approximately mimic rates observed in swine studies conducted at USAISR [24]; the hemorrhage rate was pressure-dependent, tunable in magnitude via “hemorrhage factor”, and time-dependent to mimic how the clotting factor reduces the rate. The rates for each of these are tunable to meet the desired testing requirements. In addition, penalties can be added to the hemorrhage rate, which we demonstrated as an over-pressurization penalty [28,35]. Any time pressure exceeded 2 mmHg above the threshold, the clotting factor was reset, leading to increased hemorrhage. This penalty can be tuned to be more or less aggressive, and additional penalties can be incorporated, such as for resuscitating too slowly or for using crystalloid instead of whole blood. Another element affecting the hemorrhage rate is a time-dependent simulation of coagulation that reduces hemorrhaging over time [35]. The rates for each of the hemorrhage and clotting factors are tunable to meet the desired testing requirements. Overall, the outflow logic and severity can be scenario-dependent, allowing for slow constant hemorrhage rates, hemorrhage clotting with time, or an acute massive loss of blood mimicking tourniquet failure. Combining the outflow logic with the vasopressor mechanism and PhysioVessel design, the tunability of test situations to meet end users’ needs using the hemorrhage flow loop is nearly limitless.

To better highlight the utility of the test platform, we have demonstrated an evaluation of a hemorrhage controller with different bleed scenarios in order to detail how critical controller performance metrics can be quantified [17]. Four scenarios were evaluated: (1) an initial fluid loss with a clotting hemorrhage using whole blood infusate; (2) a rapid loss of blood that stops using whole blood infusate; (3) a rapid loss of blood with bolus vasopressor delivery using whole blood infusate; and (4) rapid loss of blood with ongoing non-clotting hemorrhage using crystalloid infusate. An adapted version of the decision table logic developed by Marques et al., 2017 [11] was the demonstration controller, with the sampling rate for the controller the only variable, as development and evaluation of resuscitation controllers was beyond the scope of this work. The overall differences between sampling rates and testing scenarios were pronounced when using the developed test platform. With a lower sampling rate, the controller was prone to overshooting the target, leading to constant hemorrhage penalties, which was not observed with a more rapid sampling time. This highlights the importance of tuning controller sampling rates and sensor input rates to the hemorrhage situation. When the hemorrhage factor for certain scenarios was increased, the controller’s logic could not reach the target pressure, highlighting the need for evaluating controllers in as wide a range of testing scenarios as possible with the test platform. Lastly, the test platform allows for quantification of a range of control system performance metrics, as have been used in previous work, allowing for head-to-head comparison of controller designs and iterations [17].

While the hemorrhage test platform is robust and tunable to end users’ needs for developing and evaluating hemorrhage resuscitation systems, there are shortcomings with its application. First, the volume responsiveness of the system is set by the PhysioVessel design and cannot be adjusted in real time. This can be bypassed by using multiple PhysioVessels with different designs or by modulating the outflow rate with an additional term to indirectly adjust the responsiveness of the system. Second, the vasopressor action in its current state is static and only adjusts systemic vascular resistance. Physiologically, vasopressors increase MAP predominately by increasing systemic vascular resistance or cardiac output. Future efforts are needed in order to add cardiac output adjustments to HATRC via modulation of the stroke volume and pulse rate of the cardiovascular pump. In addition, vasopressors can be titrated to adjust the magnitude of their effect on MAP; however, this cannot be achieved without further automation of the resistance increase from vasopressor action. Third, the testing scenarios as presented were performed in succession, and the logic behind HATRC can cause carry-over effects from one scenario to the next, potentially preventing each scenario from being reviewed independently. In trauma there are often multiple injuries, and penalties are additive; nonetheless, the logic can be adjusted to ensure that a stable baseline is reset between each testing scenario. Lastly, the outflow and inflow rates selected for the project were approximated from retrospective analysis of swine hemorrhage datasets, and were not validated against other values from the literature. The goal of the presented effort was to highlight the capability of HATRC, and thus the pump rates did not need to be finalized for this initial effort. Prior to use of the system for comparing controller designs, the rates and the precise test scenarios need further validation.

The next steps for this work are three-fold. First, further automation improvements will be integrated into the test platform. While we have added solenoid valves for automated PhysioVessel swap-over in recent work [17], the vasopressor aspect has not yet been automated. A more controlled setup is needed for tuning the vasopressor effect before conducting hardware-in-the-loop closed loop vasopressor controller testing. Second, in addition to a recent study evaluating conventional controller types with HATRC, the system will be used for evaluating adaptive controllers [9,18] and thoroughly tuning the controller prior to translation to animal work. HATRC and its wide range of hemorrhage scenarios is ideal for thoroughly testing adaptive-style controllers. Third, the system will be further validated and updated with large-scale animal hemorrhage resuscitation experiments. Other future work could involve fitting HATRC to human resuscitation data in order to assist with clinical translation of controllers. In summary, HATRC as presented here allows for evaluation of closed-loop controllers with a range of testing scenarios, which can streamline development and optimization of controllers and accelerate translation for combat casualty care applications.

## 5. Conclusions

In conclusion, the hardware-in-the-loop automated testing platform presented here addresses a current gap in the development of medical devices such as closed-loop controllers. Evaluation of various types and tunings of controllers using this test platform will minimize costly and extensive experiments involving in vivo testing or clinical trials. The tunability of this platform, along with its ability to mimic relevant medical pathologies such as resuscitation, tourniquet slippage, and vasopressor administration, allows for complex and robust testing of different controllers. The ability to assess performance metrics ensures the objective and quantifiable evaluation of controllers, and will accelerate medical device development.

## Figures and Tables

**Figure 1 bioengineering-09-00373-f001:**
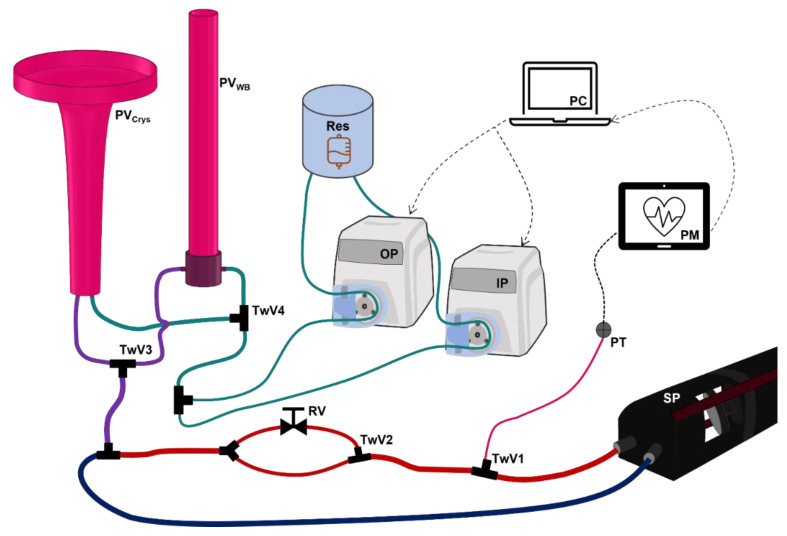
Diagram of Hardware-in-Loop Automated Testbed for Resuscitation Controllers (HATRC) Setup. An overview of the entire testbed is shown with the main circulatory flow loop; arterial and venous sides are colored red and blue, respectively. Pulsatile flow is generated by a ViVitro Labs SuperPump (SP). Then, arterial pressure is measured by a pressure transducer (PT) through fluidic connection at a three-way valve (TwV1). Arterial waveform data are displayed with a patient monitor (PM) and recorded in real-time by a computer (PC) running the hemorrhage scenarios and infusion controller. An instantaneous increase in pressure can be applied using TwV2 to direct the flow through a regulating valve (RV). Static pressure is supplied to the loop via the purple fluidic line from either whole blood (PV_WB_) or crystalloid (PV_Crys_) PhysioVessels based on the position of TwV3. Infusion (IP) and outflow (OP) peristaltic pumps are supplied with water from an external reservoir (Res) and are connected via the teal fluidic line to add or remove volume from the appropriate PhysioVessel based on the position of TwV4. Both peristaltic pumps are fully controlled by an algorithm running on the PC.

**Figure 2 bioengineering-09-00373-f002:**
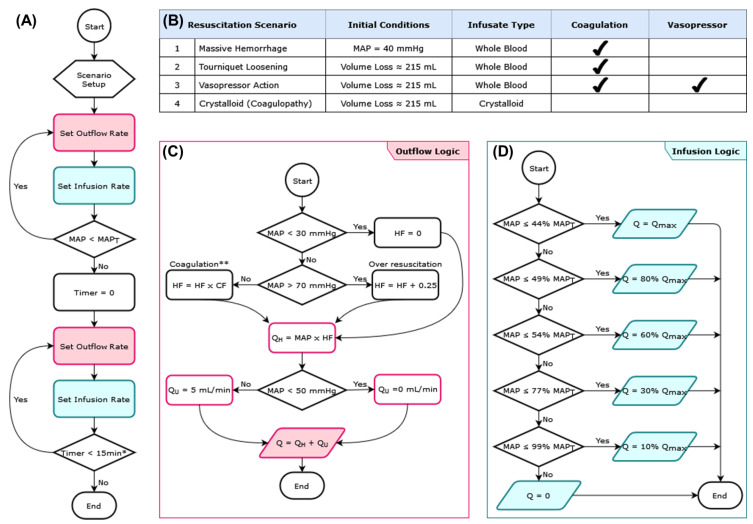
Logical components driving the HATRC platform. (**A**) Sequence of steps in the execution of each scenario. (**B**) List of simulated scenarios used to evaluate resuscitation controllers and features chosen for each. (**C**) Logic for calculating the rate for the outflow pump. (**D**) Logic for calculating the rate for the infusion pump; this decision table logic was adapted from Marques et al., 2017 [11]. * The timer logic for the fourth scenario was modified to allow it to run for 30 min. ** The “Coagulation” logic was disabled for the fourth scenario. MAP = mean arterial pressure; MAP_T_ = target MAP; Q = pump rate; Q_max_ = maximum pump rate; Q_H_ = hemorrhage rate; Q_U_ = urine rate; HF = hemorrhage factor; CF = coagulation factor.

**Figure 3 bioengineering-09-00373-f003:**
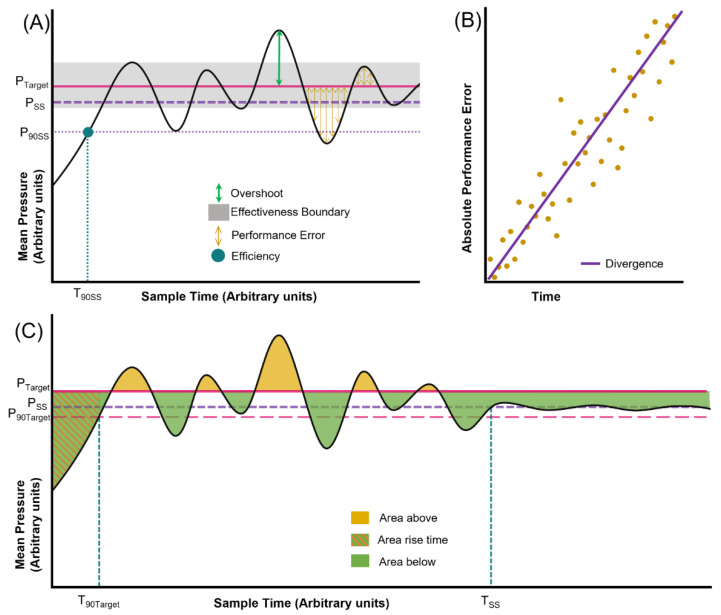
Graphical representation of controller performance metrics assessed with HATRC. (**A**) Representative mean pressure vs. time plot showing overshoot (maximum deviation from the pressure at steady state), effectiveness (boundary for the time spent within ±5 mmHg from the pressure setpoint), performance error (percent difference between MAP and setpoint), and efficiency (time required for MAP to reach 90% of the steady-state value). (**B**) Divergence is calculated as the slope of the linear regression of the absolute performance error against time. (**C**) Representative mean pressure vs. time plot showing area above setpoint pressure, area below setpoint pressure, and area rise time. Each is defined as the total area enclosed by the mean pressure curve and the setpoint line either above or below setpoint, with the area rise time limited to the region from the first measurement until the mean pressure reaches 90% of the setpoint.

**Figure 4 bioengineering-09-00373-f004:**
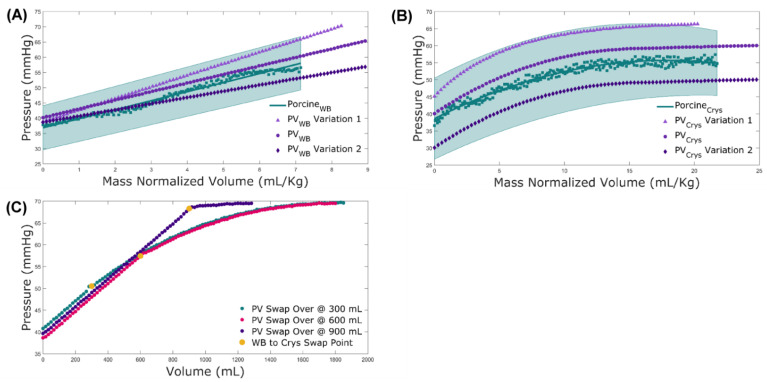
PhysioVessel Characterization. Pressure-volume testing results for (**A**) WB and (**B**) crystalloid PhysioVessels. Results are shown for base models and two PhysioVessel variations to highlight how subject variability can be introduced. Corresponding infusion data from swine for each infusate type are plotted with the regression curve fit of the data and a ±1 SD shaded region. (**C**) Switch-over test results with a PV_WB_ infusion of 300 mL, 600 mL, or 900 mL followed by a transition to PV_Crys_ until reaching target pressure (68 mmHg). Transition points between the two vessels are identified by yellow markers.

**Figure 5 bioengineering-09-00373-f005:**
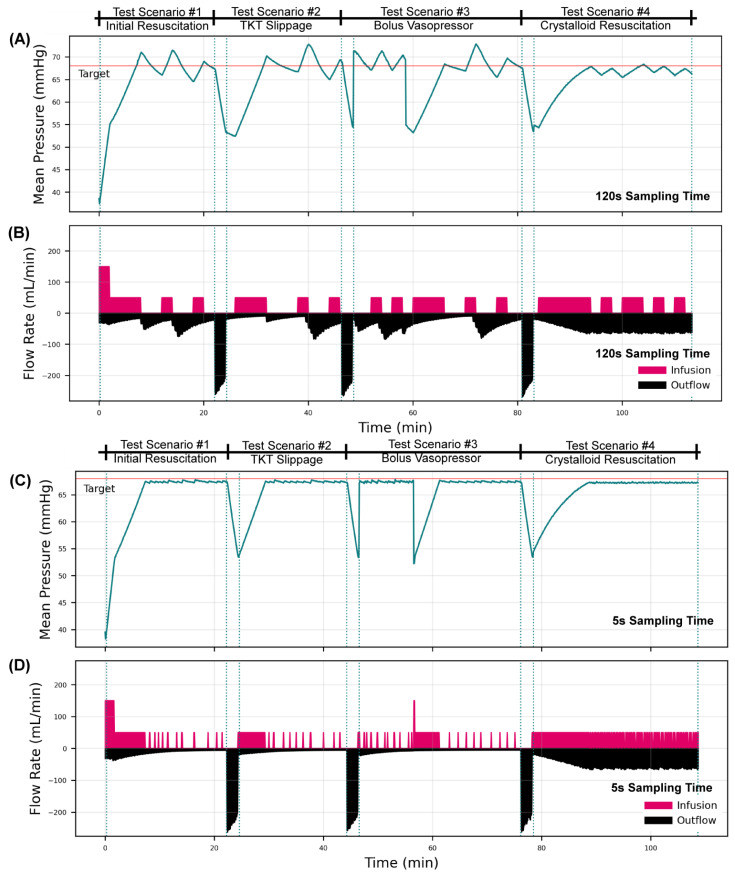
Decision Table Resuscitation Controller Performance with a low hemorrhage rate in the automated hemorrhage test platform. Testing results are shown for a decision table controller configured with (**A**,**B**) a 120 s sampling rate or (**C**,**D**) a 5 s sampling rate for receiving pressure signal and sending infusion pump rate commands. Results are shown for (**A**,**C**) the mean pressure of the system and (**B**,**D**) the infusion and outflow (hemorrhage + urine flow rates) of the system. The overall testing scheme was broken up into four scenarios, which begin at the first dotted vertical lines and end at the next dotted line, repeated for each scenario. The scenarios were as follows: (i) an initial hemorrhage to decrease MAP to 40 mmHg with ongoing hemorrhage clotting with time; (ii) a severe and rapid loss of blood, mimicking tourniquet failure; (iii) similar rapid blood loss followed by bolus vasopressor delivery lasting for 10 min; and (iv) rapid blood loss with a non-clotting hemorrhage and resuscitation with crystalloid. Low bleed corresponds to slower initial bleed rates in the first phase and continuous bleed in phase four. See HATRC Experimental Setup and Hemorrhage Test Scenarios methods for more information on the bleed logic and testing setup.

**Figure 6 bioengineering-09-00373-f006:**
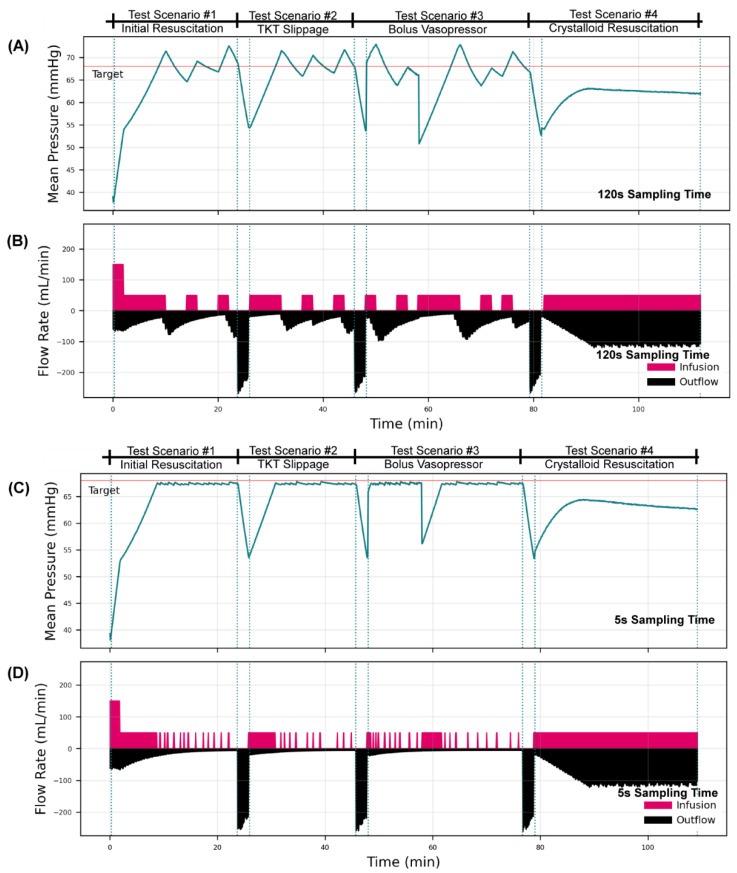
Decision Table Resuscitation Controller Performance with a high hemorrhage rate in the automated hemorrhage test platform. Testing results are shown for a decision table controller configured with (**A**,**B**) a 120 s sampling rate and (**C**,**D**) a 5 s sampling rate for receiving pressure signals and sending infusion pump rate commands. Results are shown for (**A**,**C**) the mean pressure of the system and (**B**,**D**) the infusion and outflow (hemorrhage + urine flow rates) of the system. The overall testing scheme was broken up into four scenarios, which begin at the first dotted vertical lines and end at the next dotted line, repeated for each scenario. The scenarios were as follows: (i) an initial hemorrhage to decrease MAP to 40 mmHg, with ongoing hemorrhage clotting over time; (ii) a severe and rapid loss of blood, mimicking tourniquet failure; (iii) similar rapid blood loss followed by bolus vasopressor delivery lasting for 10 min; and (iv) rapid blood loss with a non-clotting hemorrhage and resuscitation with crystalloid. High bleed corresponds to a faster initial bleed rate in the first phase and to continuous bleed in phase four. See HATRC Experimental Setup and Hemorrhage Test Scenarios methods for more information on the bleed logic and testing setup.

**Figure 7 bioengineering-09-00373-f007:**
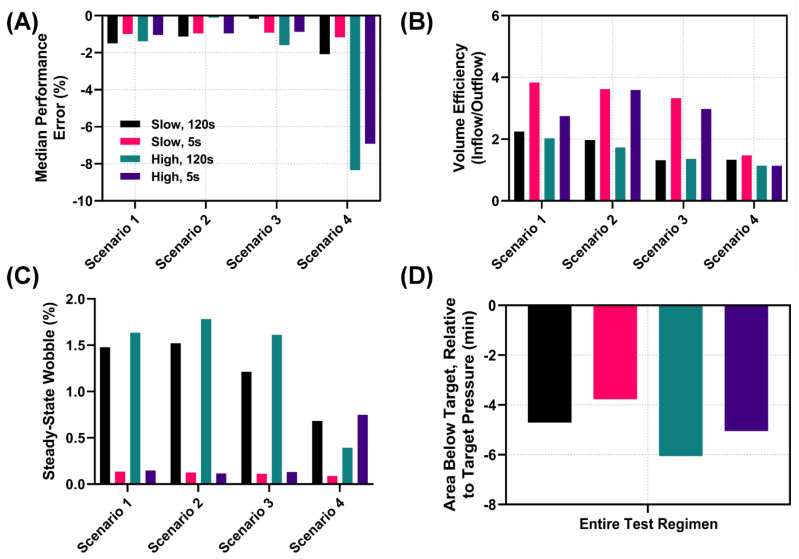
Comparison of selected performance metrics for each controller and testing configuration. Measurement of (**A**) Median Performance Error, (**B**) Volume Efficiency, (**C**) Steady-State Wobble, and (**D**) Area below the target pressure for low and high bleed rates, paired with a decision table for 120 s and 5 s sampling rates. Panels (**A**–**C**) show study results for each of four testing scenarios: (i) an initial hemorrhage to 40 mmHg MAP with ongoing hemorrhage clotting over time; (ii) a severe and rapid loss of blood, mimicking tourniquet failure; (iii) similar rapid blood loss followed by bolus vasopressor delivery lasting for 10 min; and (iv) rapid blood loss with a non-clotting hemorrhage and resuscitation with crystalloid. Panel (**D**) summarizes the entire multiple-scenario test regimen. Summary tables of additional performance metrics for each scenario and across the entire testing window are shown in the Appendix A.

## Data Availability

The datasets generated during and/or analyzed during the current study are available from the corresponding author upon reasonable request.

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
