# Peer review of "An Automated Hardware-in-Loop Testbed for Evaluating Hemorrhagic Shock Resuscitation Controllers"

_bioengineering, 2022, doi:10.3390/bioengineering9080373_

Round 1

Reviewer 1 Report

Hardware-in-Loop Automated Testbed for Evaluating Hemorrhagic Shock Fluid Resuscitation Controllers

This paper presents the development, implementation, and application of a hardware-in-the-loop system for testing of closed-loop fluid resuscitation controllers.  The paper has significance and timely in that there is an increasing interest in developing time- and cost-effective means for next-generation physiological closed-loop controlled medical devices.  The potential impact of the paper can be high.

The results presented indicate that the system is able to perform testing of closed-loop fluid resuscitation controllers and generate a multitude of important performance metrics, and also to compare multiple closed-loop fluid resuscitation controllers.

Overall, the paper is well written.  I have the following comments for the authors to consider:

1. The authors are encouraged to discuss existing efforts for hardware-in-the-loop simulation of closed-loop fluid resuscitation controllers and articulate advantages and disadvantages of the authors’ approach relative to the existing approaches.  For example:

H. Mirinejad, B. Parvinian, M Ricks, Y. Zhang, S. Weininger, J.O. Hahn, C.G. Scully, “Evaluation of Fluid Resuscitation Control Algorithms via a Hardware-in-the-Loop Test Bed,” IEEE Transactions on Biomedical Engineering, Vol. 67, No. 2, pp. 471-481, February 2020.

2. It would be beneficial to potential readers if the authors could provide more details of how the developed system can replicate dynamic 3rd spacing.  The authors mentioned in the paper (Discussion) that 3rd spacing was accounted for in the design of the PhysioVessels with Fig. 4, but Fig. 4 is a static result.

3. The system may become even more powerful if hemodynamic variables other than BP can be acquired (e.g., blood volume, hematocrit, cardiac output and total peripheral resistance).  I understand that cardiac output and total peripheral resistance can be modulated by the system, but it is not clear if physiologically plausible values of cardiac output and total peripheral resistance can be measured, preferably in real time, from the system.

4. Figure 3(B): The Y axis label may be absolute PE not median absolute PE.

5. Re: the area to setpoint metric, I wonder how generally the BP error can be mapped closely to oxygen debt and risk of re-bleeding as the authors suggested.

6. In Introduction, the authors mentioned [15] as an example of PID controller.  But, [15] actually concerns adaptive controllers.  The authors may also want to consider citing the following related paper relevant to adaptive control of fluid resuscitation:

M. Alsalti, A. Tivay, X. Jin, G.C. Kramer, J.O. Hahn, “Design and In Silico Evaluation of a Closed-Loop Hemorrhage Resuscitation Algorithm with Blood Pressure as Controlled Variable,” ASME Journal of Dynamic Systems, Measurement and Control, Vol.144, No. 2, Article Number 021001, February 2022.

Reviewer 2 Report

With a great interest I have read the work from Snider et al. on hardware-in-Loop automated testbed for evaluating hemorrhagic shock fluid resuscitation controllers. I would recommend following adapations:

Title could be revised: It is rather confusing

Line 41: please reformulate. A blood pressure of MAP 30mmHg needs to be elevated otherwise death or ischemic damage will occur.

Line 45: Please provide reference.

Line 71 “for fully automated and robust device testing prior to large animal testing.” Please reformulate

Line 79: Wouldn’t you prefer use of Packed red blood cells to the use of whole blood as blood product substitution? Please provide this as limitation in limitations section, as you developed device for WB.

What are the limitations of developed hardware? Is there any danger for potential patient?

Limitations section is missing in this paper. an should be provided.
